# The Impact of the Density of Individual Social Networks on WeChat Usage in Intimate Relationships among Chinese Youngsters

**Zhou Nie** [1,*] , **Mingzhu Li** [2] , **Moniza Waheed** [1] , **Diyana Kasimon** [1] and **Wan Anita Binti Wan Abas** [1]

[1] Department of Communication, University Putra Malaysia, Serdang 43400, Malaysia
[2] Department of Journalism and Broadcasting, Zhejiang Shuren University, Hangzhou 310009, China
[*] Correspondence: oganie728@gmail.com

**Abstract:** WeChat has become the most popular type of social media among youngsters in China. They use it for various reasons including communicating in intimate relationships. This study aims to investigate the impact of the density of individuals' social networks on WeChat Usage in Intimate Relationships among Chinese youngsters, guided by the Theory of Planned Behaviour (TPB). An online questionnaire was constructed and disseminated to respondents online. In total, 923 undergraduate students from Chinese universities completed the questionnaires. Utilizing Structural Equation Modelling, findings show that the density of individuals' social networks has a limited impact on WeChat usage. On the other hand, TPB factors such as subjective norms and perceived control bring a substantial impact on WeChat usage, while attitude has a less significant impact. These results indicate that Chinese youngsters exhibit strong attributes of the collective culture. This study also suggests that future social media research should place more emphasis on cultural and social factors.

**Keywords:** the density of individual social networks; WeChat; intimate relationships; theory of planned behavior; social media usage

## 1. Introduction

Social media enables users to create and share information and is widely used by people around the world, particularly among youngsters. In the context of this work, youngsters refer to people who were born between 1990 and 2010. They have grown up with the internet and portable digital devices, which make them digital natives. According to a global survey, youngsters have the highest social media usage rate, which was 90% in the year 2022 (GWI 2022). They have been reported to spend more than 3 h per day on average on social media, which is nearly twice as much as other generations.

The interaction patterns of youngsters on social media have caught a lot of academic attention, for their heavy social media use could exhibit how social media impact human behavior. On the aspect of dyadic interaction on social media such as intimate relationships, youngsters seek online dating at a high rate of 41% (WeAreSocial 2022). Youngsters' mental health has been heavily influenced by social media, particularly in terms of intimate relationship communication (Toma and Choi 2015; Barker et al. 2018; Dobson et al. 2018). Several studies have shown that personal relationships are not as private as people imagine, and they are often shaped by cultural and social factors (Dobson et al. 2018; Hofstede 2006). Previous research has found that social media plays a role in forming intimate relationships by providing normative concepts such as politics, social belonging, relationality, and intimacy (Plummer 2003; Dobson et al. 2018). These culturally defined social norms dictate how intimate relationships are established; thus, youngsters' intimate interactions on social media vary by culture. In this study, an investigation of social media usage among Chinese youngsters regarding intimate relationships shows how young people could be influenced

by these social norms. The social media chosen by this study is WeChat, which is reported to be the most frequently used social media among Chinese youngsters (WeAreSocial 2022).

WeChat, among all social media platforms, is the most popular social media in China. WeChat was created in the year 2011 as a platform for multi-purpose communications. It incorporates many kinds of communication patterns, such as private messages, video chatting, status posting, and comments-and-likes. These functions represent different levels of publicity, from private usage of private messages to partial publicity of group chatting, and to all publicity of full openness (Mansoor 2021). For example, people can choose different levels of publicity to allow people in different social relationships to see their posting content on WeChat, so the usage of WeChat can demonstrate the degree of interaction between people and other social relationships.

Previous research shows that people's various communication patterns are essentially determined by their social relationships. Through different social relationships, people acquire different social identities and apply different communication patterns. For example, when people are in the workplace, communication patterns with co-workers can differ a lot from communication patterns with family members. Interaction within different social relationships shapes different individual social networks, which refers to the sum of social relationships that surround a person when he or she is under a specific circumstance (Kohler et al. 2001).

People's communication patterns are influenced by individual social networks, as well as the structural features of individual social networks such as density, size, and diversity. The density of individual social networks refers to the average strength of connections around a single person, which can impact the person's behaviors by providing constraints or support (Granovetter 1983; Kohler et al. 2001). For example, the density of individual social networks can influence the chance that two persons engage in or withdraw from a romantic relationship; a dense individual social network can also encourage couples to adopt traditional social norms in their roles as husband or wife (Bott and Gluckman n.d.; Felmlee and Faris 2013; Ajzen and Fishbein 1977). These features of individual social networks have received minimal attention in the literature on social media research. As social network research reveals, individual social networks rooted in diverse social environments influence people's behaviors just as much as personal qualities like attitude, intention, and normative conceptions do (Knoke and Yang 2019).

Social media, as a hybrid platform of personal and social factors, can provide a lens to examine the impact of individual social networks on social media usage. As a result, the current study investigates the impact of the density of individual social networks on personal WeChat usage in the context of intimate relationships among Chinese youngsters, which can contribute to providing a social relationship perspective on social media research. Furthermore, this study can also contribute to giving a better understanding of the way Chinese youngsters use social media.

## 2. Literature Review

### 2.1. Intimate Relationships on Social Media and the Determinants of Social Media Uses

In this study, the main scenario is intimate relationship communication on WeChat. For the present study, an intimate relationship is defined as a romantic partnership between two people of the opposite sex that entails frequent interaction, commitment, self-disclosure, reliance, a high degree of trust, compassion, and understanding. It is generally believed that an intimate relationship is the most private interaction since it usually includes personal lives and sentiments (Dobson et al. 2018). In social media, intimate relationship communication has been transcribed into digital data, and the use traces on social media. The interaction between intimate relationship partners and the algorithms of social media has attracted a lot of intention among scholars, as the cultural backgrounds of users can be recorded by social media, which can provide a lot of evidence on how couples connect to other social support, form social capital, change ethics, and labor to create interactive

content for social media to attract other users (Dobson et al. 2018; Jamieson 2011; Kushin and Yamamoto 2013).

To understand the use of social media in intimate relationship communication, the determinants of social media uses need to be studied. Studies which include the determinants of various behaviors have caught a lot of attention from psychologists and sociologists (Yadav and Pathak 2017; Lin and Roberts 2020; Dobson et al. 2018). There are two main perspectives among these studies, one that believes that individual factors determine behavior, and the other that believes that social factors determine behavior. On the aspect of individual factors, the main point is that the psychological processing of available information mediates the effects of biological and environmental factors on behavior; that the crucial determinant of behavior is the fully functioning individual (Ajzen 2011). Among the most popular theories which hold this point is the Theory of Planned Behavior (TPB) (Ajzen and Madden 1986). TPB has gained empirical evidence to support it. This theory proposes that behavioral intention is the immediate determinant of behavior, and it mediates the effects of other factors on behavior.

On the other hand, sociologists have argued that the salient determinants of behavior are social factors such as formal and informal social networks. Different social networks provide different social norms for individuals, and these norms can facilitate or hinder individual behaviors. One person can have different behavior patterns in different social networks, and the factors which can impact behavior may change (Granovetter 1983; Wasserman and Faust 1994; Kohler et al. 2001). These studies on social networks focus on the relational nature of human behavior (Felmlee and Faris 2013), and the bridging function of social networks from the micro level of individuals to the macro level of other social groups (Granovetter 1983; Felmlee and Faris 2013).

On social media, the boundaries between individual space and public space are becoming blurry (Kushin and Yamamoto 2013; Zhang and Pentina 2012). Interactions on social media can be impacted by both individual and social factors at the same time. Therefore, regarding social media usage, a research approach that combines the individual level and social level could be needed to explain social media usage. In light of studies on social networks which hold the point that social networks could bridge the individual level and social level, this study looks at the density of individual social networks to investigate WeChat usage regarding intimate relationships among Chinese youngsters. The main aim is to examine the extent to which individual social networks impact individuals' behaviors while individuals have interactions regarding intimate relationships on social media. Thus, the combination of the individual level and social level of behavioral study forms the conceptual framework of this study. This combination of individual and social factors also reflects the feature of social media as an intermingled platform, which reflects the actual behavior of social media usage among Chinese youngsters.

Among studies on human behavior, the theory of planned behavior (TPB) has outstanding achievements in explaining different behaviors. Its framework allows the researcher to use it in contexts where more than just personal factors are present; therefore, this study will use the framework of TPB to explore the interaction of personal and social factors.

### 2.2. The Theory of Planned Behavior

The theory of planned behavior (TPB) has gained a lot of empirical achievements in situations where human behaviors are impacted by more social factors. TPB has been applied in many fields since its inception. This includes research in dental hygiene (Hoogstraten et al. 1985), education (Fredricks and Dossett 1983), moral behavior in sports (Vallerand et al. 1992b), and digital intimate relationships on social media (Dobson et al. 2018; Milardo and Allan 1997). To date, many empirical studies have proven that the TPB model has a great ability in explaining various kinds of behaviors on both individual and social levels.

Given a large amount of empirical evidence to support TPB, the variables in the TPB model were also used as references in this study. The most important determinant of human behavior is behavioral intention. Behavioral intention is the individual factor

that determines the performance of specific behaviors (Ajzen and Madden 1986; Madden et al. 1992). This variable is proposed as the only one through which all other variables can influence the behavior. Behavioral intention is perceived as the main feature of fully-functioning individuals, who can decide whether to take action as an independent person (Ajzen and Madden 1986). Previous studies show that the stronger the intention is, the more likely the behavior is to take place (Hoogstraten et al. 1985; Madden et al. 1992). There are three variables that can impact intention: attitude, subjective norms, and perceived control.

Firstly, attitude is an individual factor that reflects personal beliefs about specific behaviors and the consequences of those behaviors. The stronger the attitude is, the more intention to perform the behaviors. Many studies have proved a strong positive relationship between attitude and behavioral intention, such as consumer behavior (Ryan and Bonfield 1980), health behavior, and so on. Thus, in this study, the following is predicted:

**H1.** *Attitude has a positive influence on the behavioral intention to use WeChat in intimate relationships.*

Secondly, there is the factor of subjective norms. It represents the beliefs about how others perceive the performance of specific behaviors, and the individuals' desire to comply with these perceptions. Subjective norms present the impact of other people on individuals regarding the execution of behaviors; according to empirical studies, behaviour such as consumer behaviors (Ryan and Bonfield 1980; Yadav and Pathak 2017; Lin and Roberts 2020) and class attendance (Ajzen and Madden 1986). Subjective norms have a positive relationship with the intention to perform behaviors, which means the stronger the subjective norms are, the more intention individuals have to perform behaviors (Yadav and Pathak 2017; Ajzen and Madden 1986). Based on this, the following is proposed:

**H2.** *Subjective norms have a positive influence on the behavioral intention to use WeChat in intimate relationships.*

Thirdly, another social factor proposed is perceived control. Perceived control refers to beliefs about the ease and difficulty of performing specific behaviors. It represents the confidence of individuals in specific behaviors and the judgment about real surroundings. A previous study shows that perceived control was tested to enhance the power in predicting behavioral intention: the more control individuals think they have, the more intention to behave (Staub et al. 1971). Thus, in this study, the following is proposed:

**H3.** *Perceived control has a positive influence on the behavioral intention to use WeChat in intimate relationships.*

These three variables (attitude, subjective norms, and perceived control) are proven to be good at the explanation of the formation of intention (Sheppard et al. 1988); however, their ability to explain the formation of behaviors is not as good as intention. The intention is hypothesized as the immediate former step of behavioral execution which represents the psychological readiness to take action. According to the empirical evidence in different domains of study (Bosnjak et al. 2020; Yuriev et al. 2020), the intention could explain the majority of variation in behavioral execution, which means this variable is good at the prediction of behaviors. Thus, in this study, the following is proposed:

**H4.** *Behavioral intention has a positive influence on the use of WeChat in intimate relationships.*

The strong ability of the theory of planned behavior (TPB) to explain the formation of behaviors has attracted a lot of researchers to apply this framework, and most of these studies added more variables to the original framework to examine the roles of new variables (Ajzen and Fishbein 2008; Yadav and Pathak 2017; Soorani and Ahmadvand 2019). In this study, the main aim is also to examine the role of a new variable in social media usage. This new variable is the density of individual social networks, which represents the structural feature of the connectedness inside individual social networks.

### 2.3. The Density of Individual Social Networks

The density of individual social networks refers to the average strength of relationships in one's social networks based on self-report (Knoke and Yang 2019). This constitutes a specific set of people situated in an individual-centered social network and the relationships among them. The graph of an individual social network is presented in Figure 1.

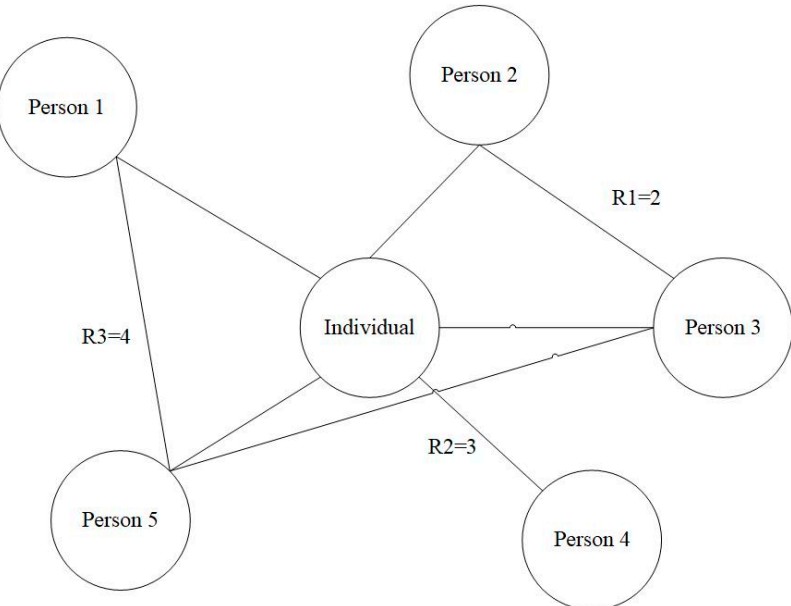

**Figure 1.** Constitution of an Individual Social Network.

As the figure above shows, in a 5-person individual social network, participants report the 5 persons they would interact with regarding specific scenarios, such as intimate relationships. Then, respondents also need to report the relationships existing among these 5 persons. The relationships are defined using a scale from 0 to 4, where 0 indicates no relationships, 1 indicates acquaintances, 2 indicates friends, 3 indicates close friends, and 4 indicates family members. In the 5-person individual social network shown in Figure 1, a respondent reported three relationships, R1, R2, and R3. R1 has a scale value of 2, indicating that R1 is a friendship relationship; R2 has a value of 3, indicating a close friendship; and R3 has a value of 4, indicating that R3 is a family relationship. The calculation of the density in an individual social network is defined as the sum of the strengths of relationships, R1 + R3 + R4 = 9, divided by the number of possible dyadic relationships, $C^N_2$. In this 5-person individual network, N = 5, thus, the possible dyadic relationships is $C^5_2$ = 5! /(2!*(5 − 2)!) = 120/12 = 10. Therefore, the density of the individual network in Figure 1 can be determined by dividing the sum of the strengths of the three reported relationships (9) by the total possible dyadic relationships (10), resulting in a density of 0.9.

As Ajzen and Fishbein (2008) suggests, new factors included in the framework of the theory of planned behavior should be proved to have an effect on behaviors. In previous studies, individual social networks were proved to have an impact on behaviors because individual social networks could provide a critical reference for people to behave. For example, couples could interact in ways that important members in individual social networks define as romantic relationships, such as interacting at a specific frequency and a certain extent of intimacy which present their couple identity (Kohler et al. 2001; Felmlee and Faris 2013). Other studies also show that individual social networks can facilitate or barricade the beginning, development, well-being, and stability of intimate relationships on social media (Sprecher and Felmlee 2000).

Specifically, as Granovetter (1983), Marsden and Friedkin (1993), and Wasserman and Faust (1994) suggest, it is not only the content of interactions in individual social networks that matters but also the structure characteristics such as size and density. There are some

studies that show that the density of individual social networks have a direct effect on intention and actual behaviors. For example, the study on fertility decision-making in the south Nyanza district in Kenya found that the density of individual social networks has a strong positive relationship with women's intention to make decisions. Because the denser the individual social networks are, the more social influence women can receive from their relatives and friends, so they are more willing to take action (Kohler et al. 2001). The density of individual social networks can also influence the participants to act or not. This is especially the case in collective behavior, where an individual's decision to behave depends on the specific proportion of people around them who make the same decision (Granovetter 1983). Considering these studies, the following hypothesis is posed:

**H5.** *The density of individual social networks has a positive influence on WeChat usage in intimate relationships.*

In other studies, the relationship between the density of individual social networks and behaviors is hypothesized to be indirect. For example, some studies claim that the impact of the density of individual social networks on behaviors was mediated by other immediate factors, such as beliefs or intentions (Granovetter 1983). Some studies have found the indirect influence of the density of individual social networks on aggressive behavior among children (Felmlee and Faris 2013). On the other hand, some studies have tested the influence of the density of individual social networks on the relationship between individual contribution and public good provision (Dijk and Van Winden 1997).

Thus, in different behavioral contexts, the density of individual social networks can have positive indirect relationships with human behaviors. As the path model presented in Figure 2 shows, the following hypothesis is posed:

**H6.** *The density of individual social networks has a positive influence on WeChat usage mainly through its influence on behavioral intention.*

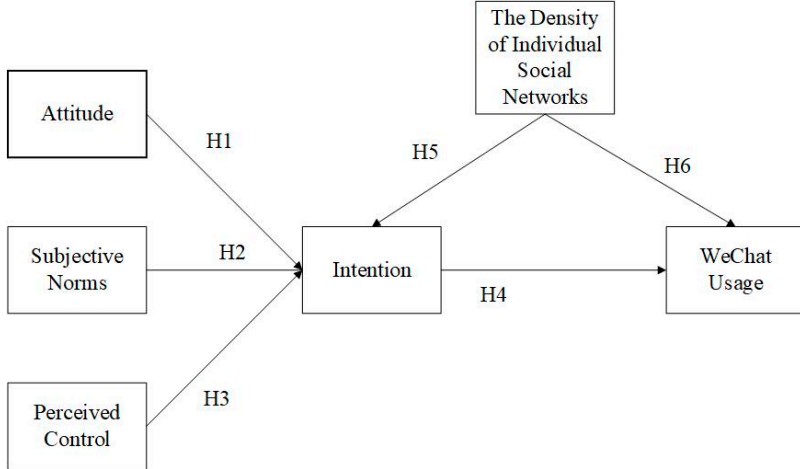

**Figure 2.** Path model for the conceptual framework of the present study.

## 3. Methodology

### 3.1. Sampling

This study's population consists of undergraduate students who have used WeChat in their intimate relationships. Because undergraduate students are young people who have grown up using digital devices, including social media, they can understand this study's questions because they have a thorough awareness of the experiences associated with using social media. This study collected data using convenience sampling (Cochran 1977; Yamane 1973), which selected respondents based on the availability of data collection, and the selection of locations is based on the data collection approvals we can get for this study. Thus, the selected universities are situated in Hangzhou city, Yunnan province, and

Guangzhou city. The sample size is 923 undergraduate students, which is composed of 57% female students and 43% male students.

### 3.2. Research Instrument

In this study, the research instrument was an online questionnaire which consisted of seven parts. The items in this questionnaire were adopted and adapted from the list of Ajzen and Fishbein (2008), because these items were also suitable for investigating social media usage. The first part is the demographic questions, which include gender, the year of study, and experience of using WeChat in intimate relationship communication.

The second part was to gauge participants' attitude. Six questions were posed. Examples of questions asked were: (1) In my intimate relationship communication, I rank my usage of (tagging/liking, commenting/message/one-to-one chatting/friend broadcasting/public conversation) as good/bad; (2) In my intimate relationship communication, I rank my usage of (tagging/liking, commenting/message/one-to-one chatting/friend broadcasting/public conversation) as beneficial/harmful; (3) In my intimate relationship communication, I rank my usage of (tagging/liking, commenting/message/one-to-one chatting/friend broadcasting/public conversation) as useful/useless. There were 6 questions to measure attitude. The first 3 were to measure behavioral beliefs, and these questions were measured with a 7-Likert scale ranging from 1 = bad/harmful/useless to 7 = good/beneficial/useful. The other 3 questions were to gauge beliefs about the consequences of WeChat usage. They were measured with a ranking system with 1 = unpleasant and 7 = pleasant.

The third part was to measure subjective norms. Six questions were asked. The first 3 questions were designed to measure the beliefs about norms that important people hold; they were: (1) Most people who are important to me approve of my use of (tagging/liking, commenting/message/one-to-one chatting/friend broadcasting/public conversation) in my intimate relationship communication; (2) My family members approve of my using (tagging/liking, commenting/message/one-to-one chatting/friend broadcasting/public conversation) in my intimate relationship communication; (3) My partner approve of my using (tagging/liking, commenting/message/one-to-one chatting/friend broadcasting/public conversation) in my intimate relationship communication. These questions were measured with a 7-Likert scale ranging from 1 = disagree to 7 = agree. The other 3 questions were to gauge the motivation of correspondents to comply with their important ones. They were measured with a ranking system with 1 = unlikely and 7 = likely.

The fourth part was to measure perceived control. There were 6 questions posed. The first 3 questions were designed to gauge beliefs about control. They were: (1) I am confident that I can use (tagging/liking, commenting/message/one-to-one chatting/friend broadcasting/public conversation) well in my intimate relationship communication; (2) For me, using (tagging/liking, commenting/message/one-to-one chatting/friend broadcasting/public conversation) in my intimate relationship communication would be easy/difficult; (3) How much control do you have over using (tagging/liking, commenting/message/one-to-one chatting/friend broadcasting/public conversation) in your intimate relationship communication? These questions were measured with a 7-Likert scale ranging from 1 = false/very difficult/absolutely no control to 7 = true/very easy/total control. The other 3 questions were to measure the pressure correspondents receive from their surroundings. They were measured with a ranking system with 1 = disagree/very few and 7 = agree/numerous.

The fifth part was about the density of the individual social network. First, participants should list 5 people with whom they are willing to share information about their intimate relationships on social media. Second, participants should label the relationship type between these people. There are 5 relationship types: 0 for no relationships, 1 for Acquaintances, 2 for Friends, 3 for close friends, and 4 for family members.

The sixth part was to gauge intention. Six questions were posed. The first 3 questions were designed to measure the readiness for performing behaviors. They were: (1) I intend to use (tagging/liking, commenting/message/one-to-one chatting/friend broadcast-

ing/public conversation) in my intimate relationship communication; (2) I will try to use (tagging/liking, commenting/message/one-to-one chatting/friend broadcasting/public conversation) in my intimate relationship communication; (3) I will make an effort to use (tagging/liking, commenting/message/one-to-one chatting/friend broadcasting/public conversation) in my intimate relationship communication. For example, I intend to use (tagging/liking, commenting/message/one-to-one chatting/friend broadcasting/public conversation) in my intimate relationship communication (unlikely 1, 2, 3, 4, 5, 6, likely 7). These questions were measured with a 7-Likert scale ranging from 1 = unlikely/not will to 7 = likely/definitely will. The other 3 questions were to measure the propensity to perform behaviors. They were measured with a ranking system from 1 = unlikely/not will to 7 = likely/definitely will.

The last part of the questionnaire was to measure the frequency of WeChat usage in intimate relationships. There were 6 questions to gauge WeChat usage. They were: (1) I use tagging/Liking in my intimate relationship communication; (2) I use commenting in my intimate relationship communication; (3) I use Message in my intimate relationship communication; (4) I use one-to-one chatting in my intimate relationship communication; (5) I use friend broadcasting in my intimate relationship communication; (6) I use public conversation in my intimate relationship communication. These questions were measured with a ranking system from 1 = barely no use to 7 = very frequent.

### 3.3. Research Procedure

The data collection was done with the help of a digital questionnaire. The targeted population of this study need to access the website to fill in the digital questionnaire. In the beginning, the targeted population was briefed about the main topic of this study, as well as information about voluntariness. After clicking on the consent button, the filling procedure began, which took about 5–9 min to finish. When the targeted population finished the questionnaire, the data was recorded on the questionnaire website; only the researchers could access the data collected.

In total, 923 questionnaires were collected, and all the questionnaires were valid responses, with no data missing. The 923 questionnaires of this study have met the prior conditions to be used in statistical analysis. Researchers organized and analysed the data after the data collection was completed. The personal information of participants in this study was completely anonymous.

### 3.4. Pilot Study

Before the formal distribution of the online questionnaire, a pilot study was done at a Chinese university located in Yunnan Province. A total of 30 students were involved in the pilot study, and the aim of the pilot study was to see the content and face validity of the questionnaire.

For the questionnaire of this study, there are 7 parts, each part was tested for the reliability, the Cronbach's α of part 2 of attitude was 0.75, part 3 of subjective norms was 0.94, part 4 on perceived control was 0.72, and part 6 about behavioral intention was 0.97. For part 5 on the density of individual social networks and part 7 on WeChat usage, test-retest reliability was adopted to see the correlation coefficient; the higher the correlation coefficient is, the higher the reliability and stability are. The time span between the test and retest was 2 months, and the correlation coefficient of the density of individual social networks was 0.80, and 0.87 for the factor of WeChat use-frequency. In the pilot study of the questionnaire, all the parts were considered acceptable to collect data in the formal survey, because all reliability indicators were above 0.70 for social science criteria.

### 3.5. Data Analysis

This study adopted structural equation modeling (SEM) to investigate the direct and indirect effects of the density of individual social networks on WeChat use regarding intimate

relationships among Chinese youngsters. SEM is an extension of the general linear model, which uses the covariance-based technique to investigate relationships among variables.

For its capacity to reflect complicated relationships between constructs within the conceptual framework of this study, SEM's multiple functions, including model fit tests, could provide good answers to this study's hypothesis.

## 4. Results

After the collection of digital questionnaires, 923 effective questionnaires had been collected to be analyzed in SEM. In this part, the analysis results will be presented, as well as hypothesis testing.

### 4.1. The Results of the Use of Remind-or-Only-Your-Partner-Could-See

In the use of the remind-or-only-your-partner-could-see function of WeChat, the regression weights and the significant levels of all variables are presented in Figure 3.

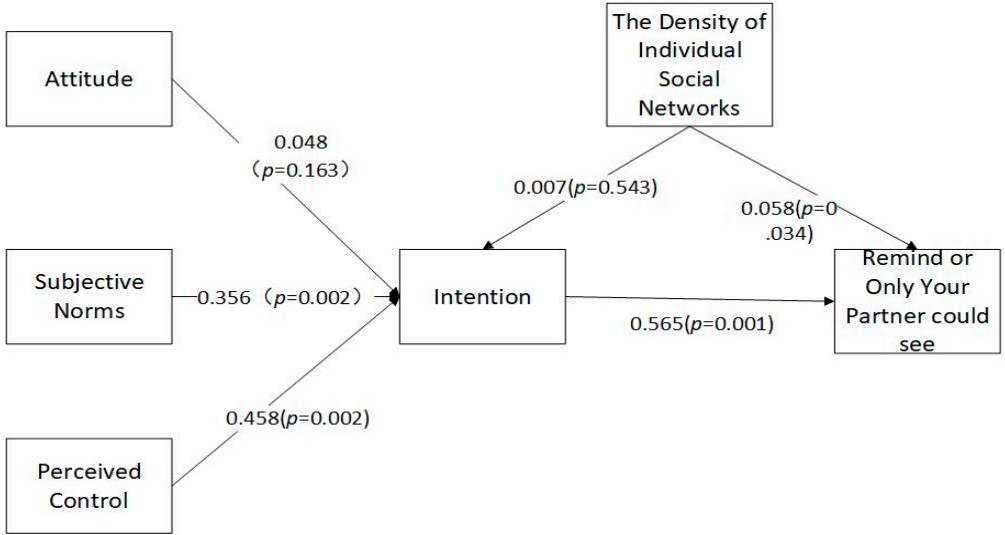

**Figure 3.** Path graph of the use of remind-or-only-your-partner-could-see.

According to Figure 3, perceived control had the greatest influence on behavioral intention, with a weight of 0.458 and a *p*-value of 0.002. While the indirect relationship between the density of individual social networks and the use of WeChat is nonexistent with a weight of 0.007 and a *p*-value of 0.543, the direct effect of the density of individual social networks on the use of remind-or-only-your-partner-could-see is weak with a weight of 0.058 and *p*-value of 0.034.

### 4.2. The Results of the Use of Private Message

The private message in WeChat is a function that enables one-to-one communication. Figure 4 displays the findings of regression weights for all factors pertaining to the use of private messages.

As shown in Figure 4, perceived control continues to have the greatest influence on behavioral intention, with a regression weight of 0.458, whereas the influence of attitude continues to be non-significant. Since the indirect relationship weight is −0.023 and the density of individual social networks has a weak regression weight of 0.054 on the use of private messages, this indicates that no indirect relationship exists between the two factors.

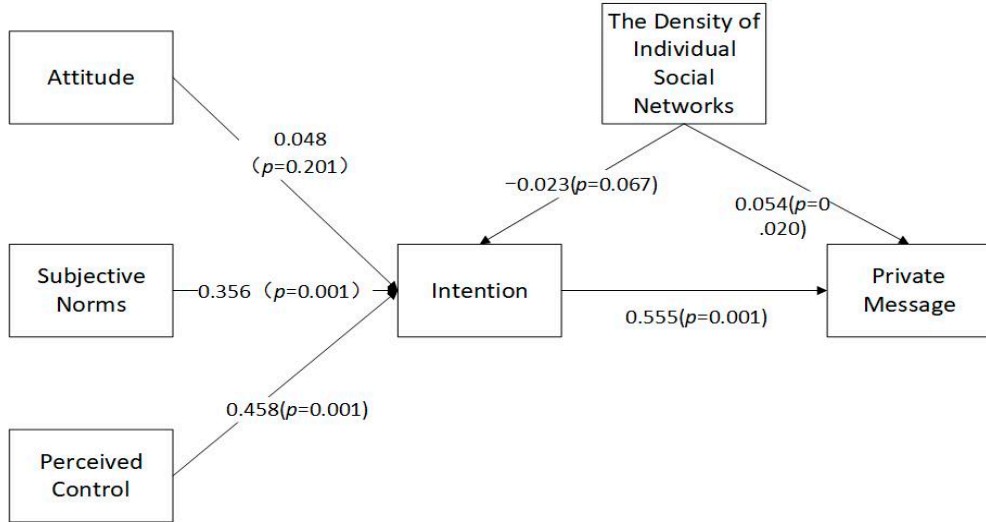

**Figure 4.** Path graph for the use of private message.

*4.3. The Results of the Use of Video-Chatting*

In this study, video-chatting in WeChat is for instant one-to-one communication. Figure 5 shows the regression weights of determinants for video chatting usage.

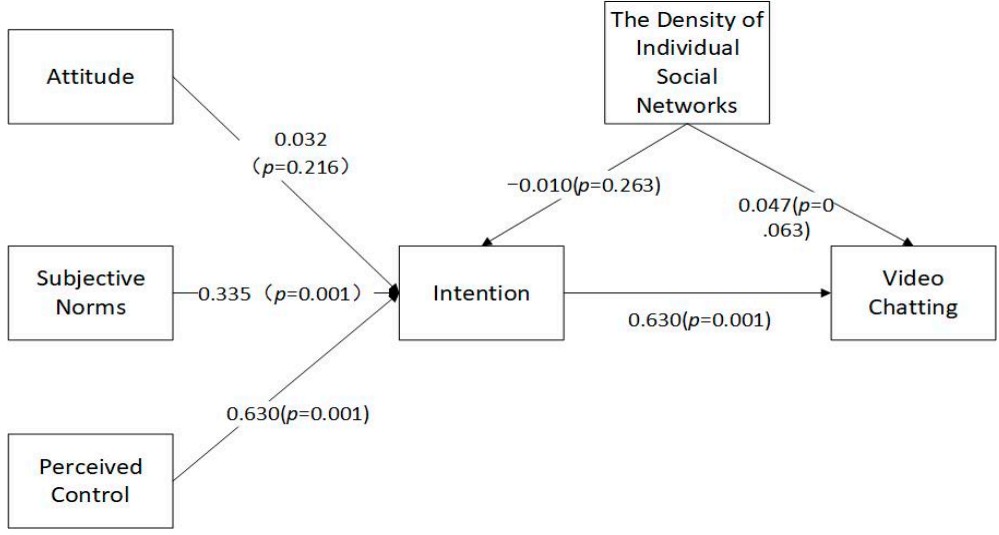

**Figure 5.** Path graph for the use of video-chatting.

Perceived control has the highest regression weight among all three antecedent variables of behavioral intention, which is 0.630. Behavioral intention is still not significantly impacted by attitude. In contrast to the strong relationship between behavioral intention and video-chatting usage, neither a direct nor an indirect correlation exists between the density of individual social networks and video-chatting usage.

*4.4. The Results of the Use of Comments-and-Likes*

Comments-and-likes in WeChat represents the reaction to the content that intimate partners post to an uncertain audience. The different impacts of variables on the use of comments-and-likes are shown in Figure 6.

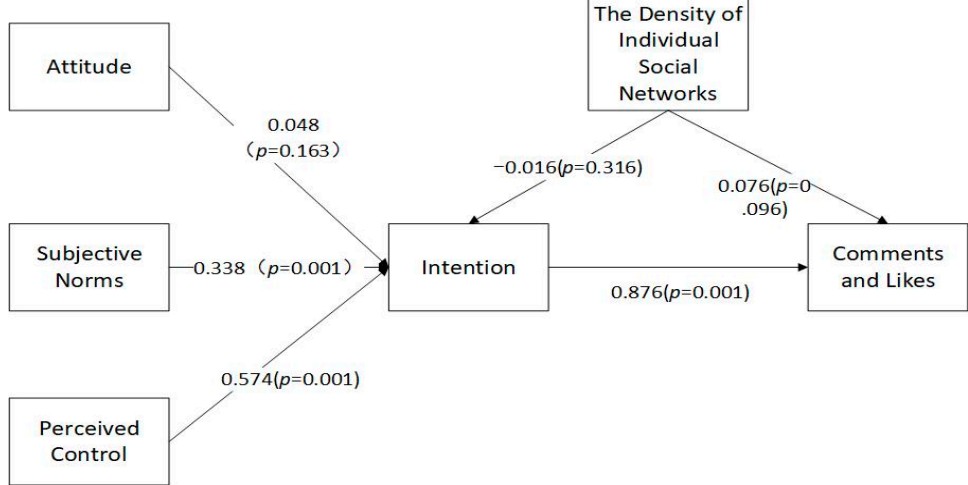

**Figure 6.** Path graph for the use of comments-and-likes.

Considering the use of comments-and-likes, perceived control has the largest regression weight of 0.574, which has the most impact on behavioral intention, while the attitude factor continues to have a non-significant influence. As behavioral intention has a very substantial relationship with the use of comments-and-likes with a regression weight of 0.876, the density of individual social networks has neither a direct nor an indirect association with the usage of comments and likes.

*4.5. The Results of the Use of Certain-Friends-Could-See*

In WeChat, the function of "certain-friends-could-see" designates semi-public communication with a selection of chosen contacts, including group-chatting and other information that could only be seen by a select number of users. Figure 7 shows the specific effects of factors on the use of certain-friends-could-see.

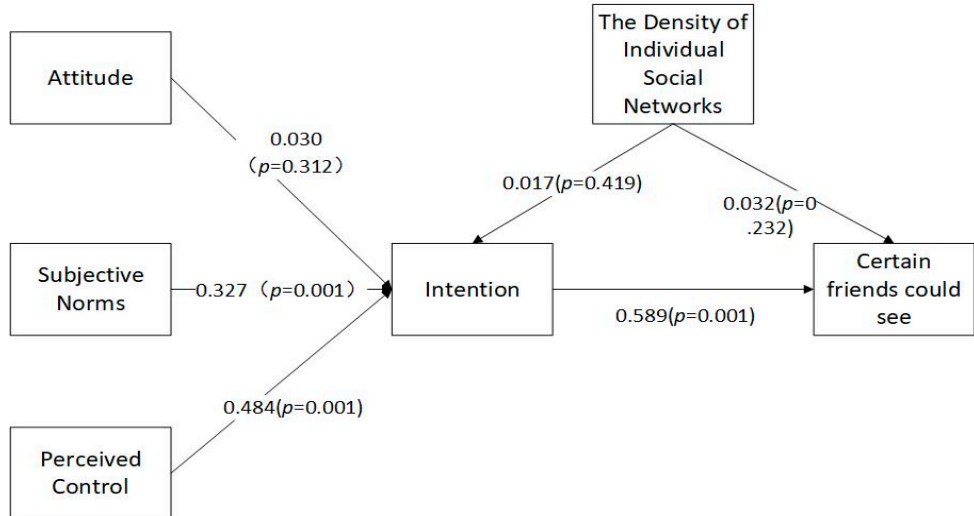

**Figure 7.** Path graph for the use of certain-friends-could-see.

According to the results, among subjective norms and attitude, perceived control has the largest weight, at 0.484. Behavioral intention is not significantly impacted by attitude. The behavioral intention has a weight of 0.589 as the immediate determinant of the use of certain-friends-could-see, meaning that it has a considerable impact on this usage. In contrast, the density of individual social networks has no direct or indirect correlations with this usage.

*4.6. The Results of the Use of All-Friends-Could-See*

When used for public communication, WeChat's all-friends-could-see function enables all contacts to view the details of communications between intimate partners. Figure 8 in this study shows the investigation into the use of this function.

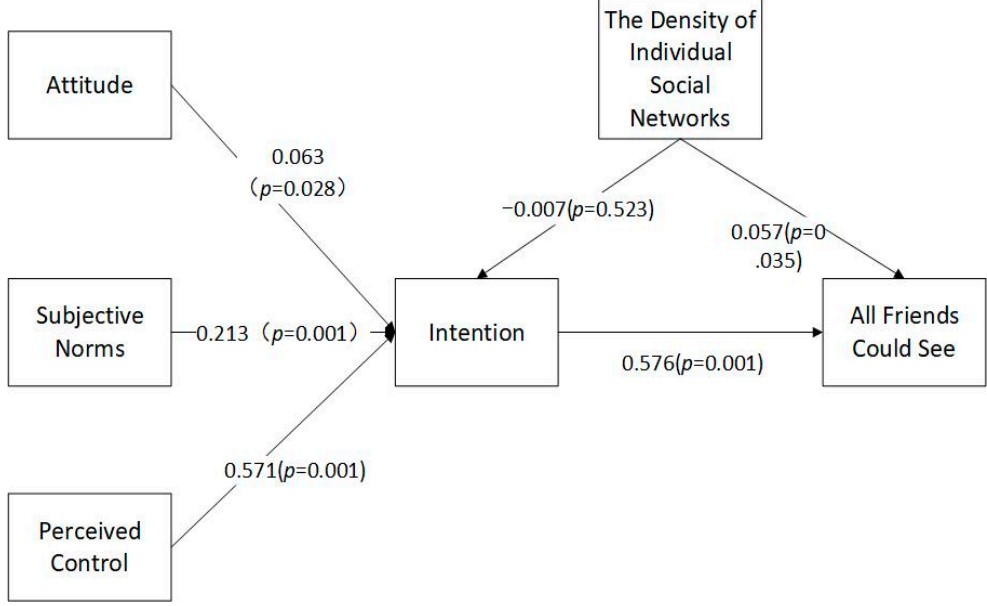

**Figure 8.** Path graph for the use of all-friends-could-see.

The findings reveal that the attitude has a weight of 0.063, which is noteworthy given that the *p*-value is 0.028 (<0.05). However, attitude has the least weight and influence on behavioral intention when compared to subjective norms and perceived control. The factor with the highest weight, perceived control, is 0.571. With a weight of 0.057 and a *p*-value of 0.035 (<0.05), the density of individual social networks has a significant direct effect on the use of all-friends-could-see, whereas the relationship between behavioral intention and use of all-friends-could-see has a much stronger weight of 0.576 and a *p*-value of .001 (<0.05).

*4.7. The Hypothesis Results*

H1 predicted that attitude has a positive impact on the behavioral intention to use WeChat in intimate relationships. Results show that attitude has almost no influence on behavioral intention. Among the 6 WeChat functions used in intimate relationships, only 1 has significant *p*-values at a 95% confidence interval, which means the variable of attitude has little influence on behavioral intention. Hence, H1 has been rejected.

Regarding H2 of this study, it hypothesized that the factor of subjective norms has a positive influence on the factor of behavioral intention. The *p*-values of all 6 WeChat usage regarding intimate relationships are significant on the level of 95% confidence interval, which are all less than 0.05. The regression weights of all 6 WeChat uses are all above 20%, which means when the factor of subjective norms goes up by 1, the behavioral intention goes up significantly. Thus, subjective norms can impact WeChat usage regarding intimate relationships significantly. As a result, H2 is supported.

In H3 of this study, the hypothesis is about the perceived control having a positive influence on behavioral intention to use WeChat. The results above show that perceived control has a significant positive influence on behavioral intentions in all 6 functions of WeChat regarding intimate relationships. Their regression weights are all positive and the *p*-values are all less than the critical value of 0.05, which means the relationships between perceived control and behavioral intentions of WeChat uses are statistically significant and positive. Thus, H3 is supported. The more control young people have, the more intention to use WeChat in intimate relationships.

H4 of this study is about behavioral intention having a positive influence on WeChat uses. As the results show above, the positive relationships between behavioral intention and WeChat usage are statistically significant with a confidence interval of 95%. The highest Beta-value is 0.876 for comments-and-likes. All Beta-values have passed 0.50, which indicates that in the model of this study, the relationship between behavioral intention and WeChat uses is statistically significant and positive. Thus, H4 is supported.

H5 is about the indirect relationships between the density of individual social networks and WeChat uses. In the test results presented above, the density of individual social networks has no indirect relationships with WeChat uses regarding intimate relationships. The *p*-values for indirect relationship tests for the density of individual social networks and WeChat uses are all bigger than the critical value of 0.05, and 0 is fallen into the range between lower and upper bounds. The influence of the density of individual social networks on WeChat uses is not mediated by behavioral intention. Hence, the hypothesis about indirect relationships between the density of social networks and WeChat uses is not supported.

H6 is about the positive relationships between the density of individual social networks and WeChat uses. The constitution of individual social networks is presented in Table 1.

**Table 1.** The density of individual social networks of Chinese young people.

| No Relations (1) (%) | Acquaintance (2) (%) | Friends (3) (%) | Close Friends (4) (%) | Family Members (5) (%) | The Density of Individual Social Networks |
|---|---|---|---|---|---|
| 77.70% | 9.97% | 3.4% | 2.3% | 7% | 2.310 |

We can see that Chinese youngsters tend to keep a low density of individual social networks when they are using WeChat for intimate relationship communication. The category of no relations has the highest percentage of 77.70%, while the mean value of the density is 2.310; this mean value has not reached half of the highest value of 5 of the density. Thus, the density of individual social networks is low regarding intimate relationships on WeChat.

As a result, the density of individual social networks has very limited influence on WeChat uses. The positive influence of the density of individual social networks exists only in relationships with remind-or-only-your-partner-could-see, private message, and all-friends-could-see. However, all regression weights are below 10% in all 6 WeChat uses. This result is consistent with other studies indicating that the lower the density is, the fewer the influence people can receive, and the low density of individual social networks can enable people to gain more control over their behavior (Lin and Roberts 2020; Consiglio et al. 2018). Thus, in this study, H6 about the density of individual social networks having a positive influence on WeChat uses is supported in WeChat uses of remind-or-only-your-partner-could-see, private message, and all-friends-could-see.

### 4.8. The Model Fit of Hypothesized Models

In the hypothesized model of this study, the density of individual social networks is added to investigate its influences on WeChat usage. To see if the new factor could fit into the framework of the theory of planned behavior (TPB), the fit indices of the overall hypothesized model are presented in Table 2.

**Table 2.** Model fit indices of the hypothesized model.

| WeChat Usage | CMIN/DF (<5.0) | CFI (>0.90) | RMSEA (<0.08) | NFI (>0.90) |
|---|---|---|---|---|
| Remind-or-only-your-partner-could-see | 7.831 | 0.992 | 0.086 | 0.991 |
| Comments-and-likes | 9.482 | 0.993 | 0.096 | 0.993 |
| Private message | 4.874 | 0.996 | 0.065 | 0.995 |
| Video-chatting | 9.882 | 0.991 | 0.098 | 0.990 |
| Certain-friends-could-see | 14.901 | 0.983 | 0.123 | 0.981 |
| All-friends-could-see | 9.450 | 0.991 | 0.096 | 0.990 |

CMIN/DF = Chi-Square value, CFI = Comparative Fit Index, RMSEA = Root-Mean-Square Error of Approximation, NFI = Normed Fit Index.

As the table above shows, in four model fit indices, compared fit index (CFI) and normed fit index (NFI) provide evidence to support model fit for hypothesized models regarding different WeChat uses. Those fit values of the default model could meet the required conditions, which are both larger than 0.90. For these two indices, CFI and NFI, the hypothesized models predict WeChat uses well.

On the other side, the other two indices show that the hypothesized models don't meet the fit requirements except for the private-message use. For that Chi-squared value (CMIN/DF) requires that the value is less than 5, while the root-mean-square error of approximation (RMSEA) requires that the value is less than 0.08. Private-message has the value of 4.874 in CMIN/DF, and 0.065 for RMSEA, which means the model of private-message is the only model that could meet all requirements of these four indices.

Thus, for the fit indices of CFI and NFI, the mode of this study is fit. However, for the fit indices of RMSEA and CMIN/DF, the model of this study is not good for the prediction of WeChat uses regarding intimate relationships.

## 5. Discussion

In the process of globalization, geographic boundaries were broken for the pursuit of productivity by improving the technique of traffic. For communication techniques, internet-based communication has weakened the boundaries of interpersonal communication and mass communication (Pettegrew and Day 2015), which has made social media an intermingled platform for individual factors and social factors (Dobson et al. 2018). These factors can impact each other more frequently and saliently than the time these factors have no chance to appear simultaneously. For example, on social media, the information coming from different social domains could together impact decision-making, such as information about purchasing coming from entrepreneurs and relatives, as well as opinion leaders. Thus, the behavior of social media usage can exhibit features of both individual and social factors.

The information explosion and weakening of boundaries bring about more freedom for people on social media (Toffler 1970). However, people tend to stick to their established cultural values when they face an enormous amount of information. This is because their cultures can provide them with familiar interpretative frameworks to comprehend situations and more feeling of control over their behavior (Bandura et al. 1980). Hofstede (2006) also pointed out that human behavior is highly determined by culture. Culture operates on a deeper level of behavior regarding emotions and memories about past experiences. The underlying cultural values cannot be changed at a level of information easily, thus, social media usage cannot change the ways people think if social media doesn't influence the deeper levels of human behavior (Hofstede 2006). Even the young generation still holds the underlying values from their cultural backgrounds. Thus, social media usage regarding intimate relationship communication could be more cultural and social, rather than global and universal (Jamieson 2011).

In this study, these two facets of social media usage can be observed by combining individual and social factors. Furthermore, there are the different levels of influence of these factors, which indicates the collective cultural backgrounds in China.

### 5.1. The Implications of the Low Density of Individual Social Networks

In this study, the relational nature of interaction on WeChat regarding intimate relationships among Chinese youngsters is discussed through the factor of the density of individual social networks. As results have shown, the density of individual social networks only has a weak direct effect on WeChat uses, and the direct effects exist in the uses of remind-or-only-your-partner-could-see, private message, and all-friends-could-see. In these use contexts, participants could receive limited influence from the density of individual social networks. However, the indications of the low density of individual social networks could be more complex than the statistical results.

Firstly, the low density of individual social networks could indicate that Chinese youngsters tend to withdraw from their social networks to spend more time with each other in intimate relationships (Huston and Burgess 1979; Johnson and Leslie 1982). Some studies pointed out that the withdrawal of couples occurs more for acquaintances and intermediate friends, instead of close friends and relatives, because couples could need support from their close relations in their intimate relationships (Sprecher and Felmlee 2000).

However, Chinese youngsters choose to withdraw from family members and close friends. This tendency could indicate that social pressure comes from close relations around Chinese youngsters. In China, the traditional Confucian disciplines for relation-building are based on conformity to close groups such as family and close friends; if people want to get recognition and the identification of the group, they need to put priority on the norms and goals of close groups rather than the needs of individuals (Hofstede 2006; Arshad and Ibrahim 2019). This could cause the suppression of individuals; thus, there is more normative pressure on Chinese youngsters rather than support when they are with close groups. The withdrawal from family members and close friends could represent the need for more control on WeChat usage regarding intimate relationships among Chinese youngsters.

Secondly, the low density of individual social networks doesn't necessarily mean less social pressure. In previous studies, it is believed that the high density of individual networks encourages couples to adopt traditional social norms and represents a high level of social pressure, while the low density of individual social networks indicates loose connections among people, which has a low level of social pressure (Knoke and Yang 2019; Granovetter 1983). However, in this study, Chinese youngsters are still heavily impacted by social norms with a low density of individual social networks. As the results show, the mean value of the reported density of individual social networks is 2.310, which is low compared with the highest value of 5. Yet the factor of subjective norms still has a strong impact on WeChat usage even though Chinese youngsters choose to withdraw from their close groups on WeChat. This could indicate that the social norms are internalized by Chinese youngsters, the social pressure could impact behaviors even when close groups are not presented. Thus, the relationship between the density of individual social networks and social pressure could be determined by cultural environments.

Thirdly, the low density of individual social networks indicates that more weak ties existed in an individual social network. As the results of this study show, strangers accounted for 77.7% of an individual social network. Weak ties could have a bridging effect that connects individuals to larger subgroups and other social organizations at the macro-level (Granovetter 1983). Weak ties are considered the channels for the diffusion of social influences, ideas, and information. Thus, more weak ties could mean more chances to be influenced by social factors, just as the results of this study show, social factors have more impact on WeChat usage than individual factors. This is also the implication for future studies which aim to study social media usage, more social factors could be considered when the individual social networks have a low density.

*5.2. The Application of the Theory of Planned Behavior in WeChat Use Regarding Intimate Relationships among Chinese Young People*

As for the factors in the framework of TPB, the factor of attitude is not a good driver for WeChat uses of Chinese youngsters regarding intimate relationships, but the factors of perceived control and subjective norms explained the most variation of WeChat uses regarding intimate relationships.

However, the factor of attitude, which is believed as a personal factor in contrast with the social factors, has no contribution to the prediction of behavioral intention and WeChat uses in intimate relationships. This situation is rare for the application of TPB, the factor of attitude normally has significant regression weight on the prediction of behavioral intention in other studies (Ajzen 2011; Vallerand et al. 1992a).

The finding of this study could exhibit some features of collective culture among Chinese youngsters. Collective cultures emphasize the needs and goals of the group as a whole over the needs and desires of each individual (Krassner et al. 2017; Baumeister et al. 2007). In this study, subjective norms are weighted a lot in the interpretation of behavioral intention and WeChat uses, which indicate the beliefs of important others and the willingness to comply with them are critical in interaction regarding intimate relationships rather than their own attitudes. This finding is typically cultural.

Perceived control could also be interpreted as a way of avoiding potential conflict by gaining more control over individual behaviors, which exhibits the feature of risk avoidance in collective cultures (Arshad and Ibrahim 2019; Hofstede 2006). Thus, the effects of factors in TPB on behaviors could vary in different social and cultural contexts, as well as their interpretation. The application of TPB should take into consideration the transience and mobility of different scenarios.

This result has been reflected in the fit indices of the hypothesized model. Only the indices of CFI and NFI show that the hypothesized model is good for predicting social media usage. This could be due to the irrelevance of the factor of the attitude and the limited effect of the density of individual social networks on social media usage. The weak prediction power of these two factors could attenuate the fit level of hypothesized model. This could indicate that the basic model of TPB may encounter more challenges in incorporating more internal and external factors which facilitate or hinder behaviors. A more comprehensive and fit research approach is needed to investigate social media usage.

*5.3. The Limitations of This Study*

For this study, there are mainly three limitations. Firstly, due to the limitation of time and resources, the sampling is limited to undergraduate students based on selected universities, so the generalization of the results to all Chinese youngsters is attenuated. Moreover, the context of this study is WeChat uses regarding intimate relationships among Chinese youngsters; other social media uses, and other cultural contexts, are not discussed in this study. However, other social media could provide different use patterns, and other cultural contexts could exhibit more differences from China. The variety of social media and cultural contexts matters because factors influencing social media usage could be very transient and dynamic. Further, social media usage is based on local experiences, thus, the universal or general approach is not enough to study various social media usage.

Secondly, the size of social networks is limited, and only the density is considered, other types of social networks, such as full social networks, are not considered. Therefore, future studies could investigate more features and types of social networks. In addition, a quantitative approach to studying individual social networks can only present the status of individual social networks, not the reasons why young people keep a low density of individual social networks regarding intimate relationships on social media. Additionally, the impacts of the socio-economic backgrounds and the levels of education on social media uses are not investigated in this study. Thus, a mixed research method including a qualitative approach could be needed to investigate individual social networks more accurately.

Thirdly, the framework of TPB could not fit enough for the inclusion of the density of individual social networks. In this study, the model fit indices show that hypothesized model has a moderate interpretation ability to explain social media uses regarding intimate relationships. A more fit research model should be needed for the inclusion of the density of individual social networks. Factors might also be identified through a confirmatory model in AMOS.

## 6. Conclusions

In some researchers' opinions, the social network represents a powerful concept for social psychology theories, for the focus on the correlational nature of human behaviors rather than the abstract social natures of behaviors (Sprecher and Felmlee 2000). In this study, the main goal was to investigate the roles of the density of individual social networks in WeChat uses regarding intimate relationships among Chinese university students. The results have shown that the density of individual social networks only has weak direct effects on the use frequencies of remind-or-only-your-partner-could-see, private message, and All-friends-could-see. This could be due to the low density of individual social networks; individual social networks in this study had a low density of 2.310, out of a total density of 5. The lower the density of individual social networks is, the lesser the influence on WeChat uses regarding intimate relationship communication. Namely, Chinese young people tend to keep their close ones away when they have intimate relationship communication on social media. The private needs for intimate relationship communication on social media indicate that Chinese youngsters want to have more control over their social media usage regarding intimate relationships.

However, this study has also provided empirical evidence that even in private interaction on social media, Chinese youngsters still have strong beliefs of social norms which are cultivated by society and the culture in which these young people were brought up. This finding could be in line with the studies of Castells (2009), which show that social media usage could be greatly adapted to suit the needs of society and the culture they are embedded in, not that society and culture make a change to fit features of social media.

Therefore, in the case of the loose density of individual social networks, Chinese youngsters feel that they have more control over the uses of WeChat in intimate relationships communication, but meanwhile, they still hold strong subjective norms which conform with their important referents, to the extent that their own attitude is not relevant in WeChat use-behaviors. This finding reveals that social media usage is highly influenced by social contexts around actants, due to the intermingled features of social media which include individuals' cultural and socio-economic background, levels of education, and digital literacy (Dobson et al. 2018). Social media usage is more contextualized than generalized, thus, more empirical studies should be done to reflect actual social media usage under different circumstances.

Furthermore, the present study indicates that the same factors in planned behavior theory could have different impacts on social media usage in different social and cultural contexts; the factors that determine behaviors could be very nuanced and contextualized. This needs more cross-cultural studies to investigate the interpretation power of planned behavior theory, hence, modification of the theory of planned behavior could be expected.

**Author Contributions:** Conceptualization, Z.N.; methodology, Z.N.; software, Z.N.; validation, M.W. and M.L.; formal analysis, Z.N.; investigation, M.L.; resources, M.L.; data curation, M.L.; writing—original draft preparation, Z.N.; writing—review and editing, M.W., D.K. and W.A.B.W.A.; supervision, M.W., D.K. and W.A.B.W.A. All authors have read and agreed to the published version of the manuscript.

**Funding:** This research received no external funding.

**Institutional Review Board Statement:** Not applicable.

**Informed Consent Statement:** Informed consent was obtained from all subjects involved in the study.

**Data Availability Statement:** Not applicable.

**Conflicts of Interest:** The authors declare no conflict of interest.

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
