# Peer review of "The Impact of the Density of Individual Social Networks on WeChat Usage in Intimate Relationships among Chinese Youngsters"

_journalmedia, doi:10.3390/journalmedia4010025_

Round 1

Reviewer 1 Report

I have to admit that I haven't read such a well-crafted manuscript in a long time. When I was reading the introduction and a concept that I did not understand appeared, it was explained in the next sentence.
The methodology is also well and clearly elaborated, which allows the study to be replicated.
I have only one remark - Ajzen and Fishbein did not write anything together in 2010. Maybe it is 2008, however this publication is missed in references.

Author Response

warm greetings!

Thank you for your precious suggestions. The relevant literature has been revised and added in references in the manuscript. Much appreciated!

Reviewer 2 Report

This study is designed to investigate the relationship between Wechat use and Intimate relationship management among Chinese young people. This study is guided by the Theory of Planned behavior. The study produced some really interesting results. However, my biggest suggestion for the authors is to create path diagram figures to display the results. It is way more convenient for the authors and more subjective to the norms of social scientific publications than using the 8 tables. Otherwise, I think the manuscript is in pretty good shape.

Author Response

Warm greetings!

Thanks for the precious suggestions! The tables in the manuscript have been replaced by path diagram figures, which has largely improved the clarity and concision of the manuscript. Much appreciated!

Reviewer 3 Report

  • This is a well-structured and well-developed paper that covers an interesting topic, with rigorous research methods and interesting research findings. However, it would be better if the authors could clarify the following points:

    • In the literature review section, it would be great if the authors could explain what they mean by intimate relationships in this paper and which intimate relationships they focus on in this study. Any existing research in this area? I think you can move some content from the discussion section to the literature review and then leave more space for the findings discussion.

    • In Page 5, the explanation of Figure 1 is quite confusing, especially when authors provide different scenarios, which are difficult to capture from the figure. For instance, consider what R2 = 3, R3 = 4, and R1 = 2, and why certain links between the individual and person are different from others. Also, it would be great if authors could provide more information on how they measure the density of the network.

    • In the methodology sampling section on page 7, the authors mention the use of convenience sampling. However, it would be great if they could provide more information on why students from Hangzhou, Yunnan, and Guangzhou are selected, and whether their location affects their social practices. Meanwhile, why only select undergraduate students as participants, Will this affect the generalisation of the research findings? Also, do authors collect any gender information or identify any gender differences in their findings?

    • In response to your discussion on the limitations of the study, maybe you can consider Dunbar's number.

    • On page 16, authors mention that the "lower the density of individual social networks is, the lesser the influence on WeChat uses regarding intimate relationship communication’, Can you provide a number here?

    • On page 17, considering the limitation of the selected sample, there is limited information on the participants’ socio-economic backgrounds and levels of education.

Author Response

Warm greetings!

Thanks for the precious suggestions! We have made the revisions according to these suggestions:

  1. The concept of intimate relationship and the type of intimate relationship of this study have been added to the literature review, as well as the literature that discusses intimate relationship communication on social media.
  2. A clearer explanation of the density of individual social networks has been added to figure 1 in the manuscript.
  3. The reason for the selection of sampling locations has been explained, which is the availability of approvals for data collection. 
  4. In the limitation section, we have added a discussion about the size and the type of social networks related to Dunbar's number.
  5. We have added a concrete number to the density of individual social networks of 2.310 in the discussion section.
  6. We have added the explanation on limitations of generalization of sampling, and the lack of discussion of the impacts of social-economic backgrounds, and level of education on social media uses.